# Testing the Resilience, Physiological Plasticity and Mechanisms Underlying Upper Temperature Limits of Antarctic Marine Ectotherms

**DOI:** 10.3390/biology13040224

**Published:** 2024-03-29

**Authors:** Simon A. Morley, Amanda E. Bates, Melody S. Clark, Elaine Fitzcharles, Rebecca Smith, Rose E. Stainthorp, Lloyd S. Peck

**Affiliations:** 1British Antarctic Survey, Natural Environment Research Council, Cambridge CB3 0ET, UK; mscl@bas.ac.uk (M.S.C.); emfi@bas.ac.uk (E.F.); rebsmi@bas.ac.uk (R.S.); rosestainthorp@gmail.com (R.E.S.); lspe@bas.ac.uk (L.S.P.); 2Department of Biology, University of Victoria, P.O. Box 1700, Victoria, BC V8W 2Y2, Canada; amandabates@uvic.ca; 3National Oceanography Centre, University of Southampton, Southampton SO14 3ZH, UK

**Keywords:** Antarctic, Southern Ocean, resilience, physiology, acclimation, acclimatisation, climate change, warming

## Abstract

**Simple Summary:**

Antarctic marine invertebrates live in the constant of the Southern Ocean and are characterised by sensitivity to small increases in temperature. We conducted a series of aquarium experiments that tested this ability and found species-specific responses to warming. We found that some species were able to survive for many months at up to 10 °C, a temperature which is up to 4 °C warmer than previously recorded. We found that the survivors of three species had adjusted their biological systems (acclimated) and were better able to survive additional rapid warming, but one anemone species did not elevate its upper temperature limit, even though it survived for 270 days at 6 °C. There were also species-specific effects of increasing oxygen concentration on long-term survival to elevated temperatures, with extended, no change, or reduced survival duration all found in different species. Thermal sensitivity is clearly the product of multiple ecological and physiological capacities, and this diversity of response needs further investigation and interpretation to improve our ability to predict future patterns of biodiversity.

**Abstract:**

Antarctic marine ectotherms live in the constant cold and are characterised by limited resilience to elevated temperature. Here we tested three of the central paradigms underlying this resilience. Firstly, we assessed the ability of eight species, from seven classes representing a range of functional groups, to survive, for 100 to 303 days, at temperatures 0 to 4 °C above previously calculated long-term temperature limits. Survivors were then tested for acclimation responses to acute warming and acclimatisation, in the field, was tested in the seastar *Odontaster validus* collected in different years, seasons and locations within Antarctica. Finally, we tested the importance of oxygen limitation in controlling upper thermal limits. We found that four of 11 species studied were able to survive for more than 245 days (245–303 days) at higher than previously recorded temperatures, between 6 and 10 °C. Only survivors of the anemone *Urticinopsis antarctica* did not acclimate CT_max_ and there was no evidence of acclimatisation in *O. validus*. We found species-specific effects of mild hyperoxia (30% oxygen) on survival duration, which was extended (two species), not changed (four species) or reduced (one species), re-enforcing that oxygen limitation is not universal in dictating thermal survival thresholds. Thermal sensitivity is clearly the product of multiple ecological and physiological capacities, and this diversity of response needs further investigation and interpretation to improve our ability to predict future patterns of biodiversity.

## 1. Introduction

Understanding the mechanisms underlying species’ resilience to environmental perturbations is key to predictions of future patterns of biodiversity. Physiological plasticity is a key element of species’ capacity to survive in fluctuating environments [1]. This plasticity consists of biochemical “resistance” pathways that can be switched on or off (e.g., anaerobic metabolism [2]; heat shock proteins [3]), but also the capacity to adjust biochemical pathways in response to altered environmental conditions [4]. Acclimation, the phenotypic response to a single stressor, often in response to laboratory-induced stress, and acclimatisation, the response to multiple stressors, often under field conditions [5], can occur over scales from hours to months. Physiological plasticity is an important component of species’ capacity, providing resilience to both short-term acute heating (e.g., heatwaves [6]) and long-term change (e.g., climate change [7]). For acclimation to be beneficial, it requires predictable “zeitgebers”, i.e., cues that are predictive of future conditions, ensuring that the cost of any physiological remodelling is outweighed by the benefit [8]. The benefit of physiological acclimation has been argued [9], as has its capacity to buffer against global warming [10,11].

Physiological responses vary over geographic and temporal scales, with many dimensions known to govern the evolution of thermal tolerance [1]. At one extreme for marine life, Antarctic marine ectotherms have evolved in the constant cold of the Southern Ocean for tens of millions of years and are predicted to be sensitive to warming, but also have limited physiological plasticity [12,13]. Antarctic marine ectotherms have atypical thermal responses, including the expression of heat shock proteins, an almost ubiquitous protection of proteins to acute warming (HSP [14]), failure to select preferred temperatures [15] and in some species, lack of escape response to dangerously high temperatures [15,16]. This combined evidence supports that universal protective responses to heat regimes experienced in the Antarctic could be missing in these taxa. Recent studies have shown that the responses of Antarctic marine ectotherms are much more complicated. For example, longer-term incubations can lead to acclimation of some Antarctic marine ectotherms [11] but not others [11,17], and cellular responses, including HSP expression, vary markedly between species and treatment regimens [18]. In the cold of the Southern Ocean all physiological processes take longer, and acclimation in Antarctic marine ectotherms can take many months and can occur more slowly in invertebrates than fish [11].

Mixed responses indicate that the mechanisms underlying thermal sensitivity are complex. Oxygen and capacity limitation of thermal tolerance (OCLTT [2]) is one of the mechanisms whose applicability has been questioned in the Antarctic [1], but also more broadly with studies finding mixed responses both inside and outside the Southern Ocean [19,20,21]. 

Here we synthesise a series of experiments conducted from 2006 to 2015 that investigated both the acclimation and acclimatisation capacity of selected Antarctic marine ectotherms. Previous estimates of upper thermal limits for the long-term survival of Antarctic marine ectotherms were extrapolated to temperatures between 1 and 6 °C for incubations of several months [12]. To test if 6 °C represents the long-term limit for survival, species from a range of phyla were selected to represent a range of functional groups, the ophiuroids *Ophionotus victoriae* (Bell, 1902) and *Ophiura crassa* (Mortensen, 1936), the echinoid *Sterechinus neumayeri* (Meissner, 1900), the asteroid *Odontaster Validus* (Koehler, 1906), the holothuroid *Heterocucumis steineni* (Ludwig, 1898), the anthozoa *Urticinopsis antarctica* (Verrill, 1922), and the molluscs *Margarella antarctica* (E. Lamy, 1906) and *Laternula elliptica* (P.P. King, 1832), were incubated at 6 and 8 °C for up to 10 months. Survivors were tested for whole body acclimation to see if acute thermal limits (CT_max_—the critical thermal maximum) had been elevated through exposure to these elevated temperatures. Acclimitisation of *O. validus* was tested by comparing CT_max_ from field fresh individuals just after collection from the shallow sea around the British Antarctic Survey’s Rothera Research Station in spring, summer and winter seasons, and McMurdo Sound near Antarctica New Zealand’s Scott Base. The aim was to test for spatio-temporal variation in CT_max_. To test if elevated oxygen increased survival times at elevated temperatures, species were incubated long-term under normoxic (21%) and mildly hyperoxic (30%) oxygen saturation. The arthropod *Paraceradocus miersi* (Pfeffer, 1888), the ophiuroids *Ophionotus victoriae* and *O. crassa* were incubated at 3 °C and *S. neumayeri*, *O. Validus*, and the holothuroids *Cucumaria georgiana* (Lampert, 1886) and *H. steineni* were reared at 8 °C rising to 10 °C, under both normoxia and mild hyperoxia. In this way, this study further investigated the paradigms of limited physiological resilience, plasticity and the importance of oxygen setting upper temperature limits in Antarctic marine ectotherms, searching for exceptions and potential new insights into the underlying mechanisms.

## 2. Materials and Methods

### 2.1. Collections and Animal Husbandry

Experiments were conducted between 2006 and 2015 with individuals hand collected by SCUBA divers in the austral summer, from 6–15 m depth, near Rothera Research Station, Adelaide Island (67°34′25″ S, 68°08′00″ W). Common marine ectotherms were selected from different phyla to represent a range of functional groups, for which aquarium husbandry is well established and long-term temperature limits have been estimated under normoxia [12]. All animals remained submerged throughout the transfer from the sea to the flow-through aquarium system at the station. To control for size-dependent effects on survival, we selected individuals of a similar size within each species group at the start of the experiment, and only studied fully reproductive adults.

The 3.0 °C temperature –oxygen experiment was conducted in flow through aquaria at Rothera Research Station. In Rothera, the tanks had a constant exchange of seawater that was balanced to allow temperature and oxygen treatments to be maintained while preventing any build-up of metabolic waste. In all experiments, seawater chemistry was monitored every 2–3 days using Nutrafin aquarium test kits. Ammonia, nitrite, and nitrates were maintained well below 0.4, 0.2 and 5 mg L^−1^ to prevent toxicity from metabolic by-products.

All other animals were transported back to the UK in purpose-built, recirculating, temperature-controlled transport aquaria maintained at 0.0 ± 0.3 °C with an average 12L:12D photoperiod for the six-week passage. Once in the UK, the animals were kept in a temperature-controlled aquaria at 0.1 ± 0.1 °C until the start of experiments. In the UK, holding and experiments were conducted in recirculating tanks that were fitted with biological filters (EHEIM GmbH & Co KG, Stuttgart, Germany), UV sterilisers, bubbled air, and plastic lids. Water quality was maintained with a combination of biological filtration, protein skimming, and partial seawater exchanges (approximately 5–15% every 2–3 days), which was confirmed as detailed above. Throughout transport and experimental incubations all animals were fed, to excess, on small pieces of fish or crustaceans except for the filter feeders *C. georgiana*, *H. steineni*, and *L. elliptica* which were fed on a mixture of instant concentrated *Nanochloropsis* spp. and *Tetraselmis* spp. (ZM Fish Food, Winchester, UK) and algae growing naturally in the water.

In all experiments, functional survival (CT_max_) was monitored daily in both treatment and control animals through the following standard procedures *cf* [22]. For all but *P. miersi*, the first sign that individuals were approaching their limit was the inability to remain attached to the side of the tank. These individuals were then tested daily for the absence of movement of arms, spines, and tube feet. For *P. miersi*, we used the absence of movement of appendages (uropods or pleopods). If an individual was non-responsive it was returned to the treatment tank and checked again 24 h later. If, after 24 h, there was still no response to external stimuli then the number of days from the start of the experiment to the first day with no response was recorded as the time taken for CT_max_ to occur. To account for any effect of size on mortality, the size of each individual was recorded.

### 2.2. Identification of Anemone Species

A small piece of one tentacle was removed from each anemone and preserved in 96% ethanol. DNA was extracted from each tentacle using the DNeasy Blood and Tissue kit (Qiagen, Manchester, UK) according to manufacturer’s instructions. The cytochrome oxidase subunit I gene (COI) mitochondrial region was amplified using 1–2 µL extracted DNA and MyTaq DNA polymerase mix (30 µL reactions; Bioline UK (now Meridien Biosceince, London, UK), with 10 nmol each of the universal COI primers for invertebrates (LCO 1490 5′-GGTCAACAAATCATAAAGATATTGG-3′; HCO 2198 5′-TAAACTTCAGGGTGACCAAAAAATCA-3′) [23]. PCR conditions were 94 °C for 5 min, five cycles of 94 °C for 1 min, 45 °C for 1.5 min, 72 °C for 1.5 min, followed by 30 cycles of 94 °C for 1 min, 50 °C for 1 min, 72 °C for 1 min and a final elongation stage of 5 min at 72 °C. The COI fragments were bi-directionally sequenced by Source Bioscience (Cambridge, UK). The species identity of each individual was analysed using Blast sequence similarity searching of INSDC (International Nucleotide Sequence Database Collaboration) (https://www.insdc.org/, accessed on 12 March 2020).

### 2.3. Acclimation Experiments: Temperature Incubations

Temperatures were raised at the same rate in all experiments (0.3 ± 0.1 °C d^−1^) until the incubation temperature was reached, and subsequently monitored daily (Appendix A). *O. victoriae*, *O. crassa*, *S. neumayeri*, *O. validus*, *H. steineni*, *U. antarctica*, *M. antarctica*, and *L. elliptica* were held at temperatures of 6.0 and 8.0 °C, temperatures that were 0 to 4 °C above the long-term limits calculated from experiments with different rates of warming [12]. Control animals were kept at 0.0 °C in the main holding aquarium.

Acute thermal limits: individuals that survived beyond the duration of incubations at both 6.0 and 8.0 °C were then tested to see if acute thermal limits were elevated due to acclimation to these elevated incubation temperatures. These data were compared with acute thermal limits conducted on control individuals that had been kept in the aquarium at 0.0 °C. For the acute temperature ramping trials individuals were transferred to plastic jacketed tanks (Engineering Design and Plastics Ltd., Cambridge, UK), whose jackets were filled with 25% *v*/*v* ethanol in water solution, that was heated or cooled by LTD20G thermocirculators (Grant Instruments, Royston, UK). Temperatures were increased at 1.0 ± 0.1 °C d^−1^ until the last animal was no longer responding to the stimuli detailed above.

### 2.4. Field Comparisons

Acclimatisation in field animals was tested for in only one species, the common starfish *O. validus*. To test for field acclimatisation the assessment of CT_max_ of freshly collected *O. validus* was repeated between 2006 and 2015 in both summer and winter. There was also a test of individuals from Scott Base (77°50′57″ S, 166°46′06″ E) to allow for regional differences and comparison with a site with even less annual and seasonal variation than Rothera. As above, temperatures were increased at 1.0 ± 0.1 °C d^−1^ until the last animal was no longer responding to the stimuli. For analysis, field temperature from the Rothera Time Series [24] was measured by CTD (conductivity, temperature, depth) casts at 15 m depth and was averaged for the month of animal collection.

### 2.5. Temperature–Oxygen Incubations

In temperature–oxygen incubations, temperatures were raised from ambient at the same rate of warming as in previous trials (0.3 ± 0.1 °C d^−1^). In the 2010/11 experiments, individuals of *P. miersi*, and the ophiuroids *O. victoriae* and *O. crassa*, were incubated at 3.0 °C (Table 1), temperatures that were 1 °C above the calculated long-term limits for *O. victoriae* and that matched those for *P. miersii*, based on the results of the acclimation experiments and calculated long-term lethal limits [12]. *S. neumayeri*, *O. validus*, *C. georgiana*, and *H. steineni* were incubated long-term at 8.0 °C. After 148 days, survival of *H. steineni*, *S. neumayeri*, and *O. validus* was greater than 95%, and to increase the chance of testing for treatment effects, the incubation temperature was raised to 10.0 °C and maintained until the end of the experiment (351 days in total).

Oxygen levels were controlled by vigorously bubbling gas mixtures with different oxygen concentrations through the water to ensure oxygen did not fall below the required saturation and that the tanks were sufficiently mixed. In the Rothera experiments, premixed cylinders with 30% oxygen, 390 ppm of CO_2_ with the balance nitrogen, were used to control hyperoxia (BOC, London, UK). In the UK experiments, cylinders of 100% oxygen and air pumps were used, with the flow of gas mixtures into each tank controlled by needle valves and adjusted as necessary to ensure that required oxygen concentrations were maintained (Table 1). Stability of the oxygen concentration in all tanks was confirmed by daily monitoring of oxygen levels during set-up of the experiment using a pre-calibrated oxygen-sensitive foil and a Fibox-3 oxygen meter. We found that the stability of dissolved oxygen concentrations was such that after set-up, monitoring was only required when a gas cylinder was changed. In each experiment, controls were set up with animals held at normoxia (21% O_2_) and a temperature of 0.0 ± 0.1 °C in the respective aquaria for the duration of the experiment with no mortality.

### 2.6. Statistical Analysis

We used the Kaplan-Meier estimator to compare survival between treatments as it takes account of individuals that survived beyond the duration of the incubation (R package survival 3.5-8 [25]). Kaplan-Meier estimates median survival time including survivors as censored observations. Mantel-Haenszel tests were used to test for differences between temperature or oxygen treatments within each species.

For analysis of acute lethal limits after acclimation, trial (as a fixed factor) and replicate (as a random factor to account for non-independence of species) were added to account for any differences between the different experiments. To account for animal size, wet weight was added as a covariate. Lethal limits were Box-Cox transformed to normalise the residuals from ANOVA analysis (or in the case of *U. antarctica*, *H. steineni*, and *O. Validus*, acclimatisation to assess them as visually approximately normal). All data sets had equal variance (Levene’s test). To test for acclimatisation, CT_max_ of *O. validus* was regressed against average water temperature during the month of collection.

## 3. Results

### 3.1. Identification of Anemone Species

COI DNA sequences identified 27 out of 32 anemones as *Urticinopsis antarctica*. Five individuals were not identified with sufficient certainty and were removed from further analysis.

### 3.2. Temperature Incubations

In six out of eight trials, species survived for significantly longer when incubated at 6.0 than 8.0 °C (Figure 1, Table 2 and Appendix A, Appendix A). The two exceptions were *M. antarctica*, whose median survival duration (13 and 17 days) was not significantly different at 6.0 and 8.0 °C (Chi^2^ = 2.4, *p* = 0.1; Appendix A), and the trial with *S. neumeyari*, when the majority of individuals survived for the duration of incubations (303 days) at both 6.0 and 8.0 °C (Chi^2^ = 3.7, *p* = 0.06; Appendix A). When considered across all trials, species for which median survival durations could be estimated at both 6.0 and 8.0 °C survived for significantly longer at the lower temperature (Chi^2^ = 24.5, *p* < 0.01). *O. validus* also survived more than 200 days at 8.0 °C until a technical failure of the tank.

### 3.3. Acclimation of Acute CT_max_

*U. antarctica* acute CT_max_ was not significantly different when incubated at either 0 or 6 °C (F_(1,25)_ = 0.7, *p* = 0.69), with no effect of animal weight (F_(1,25)_ = 0.6, *p* = 0.61) or trial (F_(1.25)_ = 1.0, *p* = 0.33; Figure 2A; Appendix A). *H. steineni* incubated at 6.0 °C had significantly higher CT_max_ (F_(1,85)_ = 8.2, *p* < 0.01) than those incubated at 0.0 °C (Figure 2B) but with no effect of animal weight (F_(1,85)_ = 0.2, *p* = 0.66; Appendix A) or trial (F_(1,85)_ = 0.8, *p* = 0.36). There was, however, a significant difference between replicates (F_(1,85)_ = 5.9, *p* = 0.02), with the first incubation having significantly higher CT_max_ (12.7 ± 0.34 °C at 0.0 °C and 13.6 ± 0.29 °C at 6.0 °C) than the second (12.1 ± 0.31 °C at 0.0 °C and 12.8 ± 0.34 °C at 6.0 °C). *O. validus* incubated at 6.0 °C had a significantly higher CT_max_ (F_(1,31)_ = 11.1, *p* < 0.01) than those incubated at 0.0 °C (Figure 2C), with no effect of animal weight (F_(1,31)_ = 3.7, *p* = 0.07; Appendix A), replicate (F_(1,31)_ = 0.1, *p* = 0.72) or trial (F_(1,31)_ = 2.2, *p* = 0.15). Acclimation of CT_max_ was also apparent for *S. neumayeri* (F_(1,50)_ = 67.7, *p* < 0.01), with no effect of animal weight (F_(1,50)_ = 1.0, *p* = 0.3) or replicate (F_(1,50)_ = 1.5, *p* = 0.23; Appendix A). Lethal limits were higher at 6.0 °C (T = 9.6, *p* < 0.01) and 8.0 °C (T = 9.4, *p* < 0.01) than 0.0 °C (Figure 2D).

There was no significant effect of average environmental (seawater) temperature during the month of collection on CT_max_ measured in *O. validus* (F_(1,264)_ = 0.82, *p* = 0.37; Figure 3).

### 3.4. Temperature–Oxygen Incubations

In the first experiment all individuals of *O. victoriae* reached CT_max_ within the 99 days at 3.0 °C. The survival duration of both *O. victoriae* (median 84 versus 66.5; Chi^2^ = 47.8, *p* < 0.01) and *O. crassa* (>100 versus 82; Chi^2^ = 41.4, *p* < 0.01) was extended under hyperoxia (30%) compared to normoxia (21%; Figure 4 and Appendix A; Appendix A). A total of 87% of *O. crassa* reached CT_max_ within 100 days under normoxia, whereas only 13% reached CT_max_ under hyperoxia. The other difference in survival duration was *O. validus* incubated at 8.0 increased to 10.0 °C, which showed the opposite pattern, and survived significantly longer under normoxia (median > 351 days) than hyperoxia (322 days). There was no significant effect of oxygen concentration on survival of the other species but when all species were considered together, survival was longer under hyperoxia (30%) than normoxia (21%; Likelihood ratio, Chi^2^ = 15.8, *p* < 0.01).

## 4. Discussion

Overall, we report mixed support for three central thermal physiology paradigms, with species-specific responses in: (1) physiological tolerance, (2) long-term acclimation, and (3) the effect of oxygen limitation on long-term survival. Previous estimates of long-term thermal tolerance of Antarctic marine ectotherms, extrapolated from CT_max_ assessments at multiple rates of warming, were between 1 and 6 °C if warming occurred over months, but dropped to a predicted average of 1.3 °C after about a year [12]. However, the current study extended known long-term incubation limits, with some individuals of *U. antarcticus* and *H. steineni* able to survive long-term at 6 °C, and *O. validus* and *S. neumayeri* at temperatures up to 10 °C (Table 1). Of all the Antarctic marine ectotherms tested to date, *O. validus* is one of the most tolerant, with the collapse of physiological processes (food processing and coelomic oxygen concentration) at 6 °C and loss of activity at 9 °C [12]. *S. neumayeri* also has a thermal tolerance towards the upper-end of the Antarctic ectotherm assemblage, with average functional thresholds in short-term warming around 8.3 ± 1.3 °C [26]. This study extended the long-term tolerance of *O. validus* and *S. neumayeri* to higher than previously reported temperatures of up to 10.0 °C, although it should be noted that individuals might have been resisting warming for long periods and might have impaired functions that would preclude, e.g., reproduction and recruitment. This is likely the case for *U. antarcticus*, which did not show evidence of acclimation, even after 245 days at 6 °C.

The current study, therefore, extends the known long-term experimental upper thermal limits of several Antarctic marine species to up to 8.2 °C, well above current maximum summer temperatures recorded in Marguerite Bay (1.8 °C [24]). The warming trend of air temperature along the Western Antarctic Peninsula since the 1970s has continued after a brief hiatus around the turn of the 21st century [27]. However, the warming signal in the Southern Ocean is less clear, with slower than average surface warming (0.02 °C per decade since 1950) [28]. The findings of the current study therefore suggest that the thermal limit of the most tolerant shallow water Antarctic marine ectotherms will not be breached for several centuries. However, thermal tolerance is species-specific, with limits for less tolerant species such as *O. victoriae* (this study and [19]), and the encrusting spirorbid worm *Protolaeospira stalagmia* [18], likely be reached in much less than 100 years, which will directly impact ecosystem biodiversity. Also, the limits for activity are often much lower than CT_max_ estimates, with species expected to suffer ecological impacts at temperatures well below CT_max_. Species such as the scallop *Adamussium colbecki*, the limpet, *Nacella concinna*, and the bivalve mollusc, *L. elliptica*, have behavioural limits between 2 and 3 °C [28] which will clearly affect long-term survival in the wild. Understanding the variation in thermal tolerance between species, and between physiological functions, is crucial for determining the likely winners and losers in response to climate change and, therefore the impact of these changes on ecosystem function.

The responses of species to the effect of elevated oxygen on long-term survival arguing against the OCLTT being a universal mechanism controlling upper temperatures limits. While some species had clearly elevated CT_max_ under mild hyperoxia, indicating a positive effect of oxygen on whole animal CT_max_, the response was not consistent across species and *O. validus* even showed a slight negative effect of hyperoxia on survival. As with the findings of the current study, Clark et al. [19] found multiple mechanisms underpinning species response when incubated at temperatures just below CT_max_, with only two species showing the switch from aerobic to anaerobic metabolism [19]. Species-specific thermal tolerances have been linked to their functional traits, with the most active species having higher upper lethal limits [12]. Activity is correlated with aerobic scope, with more active species having an oxygen supply cascade that is capable of supplying more active tissues, resulting in a greater “total excess aerobic power budget” and therefore a higher capacity to resist thermal challenge [12,29]. However, where in the past Antarctic marine ectotherm species with higher activity levels have survived warming to higher temperatures [12], and this was taken as support for OCLTT, the species that acclimated to the higher temperatures in this study were all low-activity species, arguing against the importance of oxygen limitation. In the current study, the most resistant species came from a mixture of functional groups, from a mobile obligate predator through to a sessile primary consumer. It should be noted that the use of CT_max_ to test the OCLTT has been criticised and future research should include sub-lethal assays of thermal tolerance such as activity and the response of physiological pathways [30].

High concentrations of dissolved oxygen are expected to form free radicals, which could lead to tissue damage during long-term incubations [31]. If oxygen was not limiting to *O. validus* at the chosen incubation temperatures, then mild hyperoxia could have resulted in oxygen free-radical damage or additional costs for antioxidant defence mechanisms (e.g., superoxide dismutase [31]). This could potentially explain the negative effect of hyperoxia, but also provides a clear indication that oxygen was not limiting at these temperatures for this species. It is possible that a different pattern will be seen at higher temperatures.

At temperatures typical of polar oceans, the acclimation of physiological systems of marine invertebrates can take many months, with successful whole animal acclimation being recorded in three out of eight Antarctic marine invertebrate species within 2 and 9 months at 3–8 °C ([11] and current study). Acclimation in Antarctic fish was faster, with seven out of nine species acclimating within 5 to 36 days at 4 to 7 °C [32,33,34,35,36,37]. In the current study, the anemone *U. antarcticus* survived, with minimal mortality, for 270 days at 6 °C but with no whole animal adjustment of CT_max_. Caution must be applied when interpreting the CT_max_ of survivors in thermal challenge experiments, as this will likely overestimate resilience, as only a subset of the more resilient survivors is being tested (5–95% survival). Also, the results under carefully controlled laboratory conditions may not reflect survival in the wild when sub-lethal effects will likely limit long-term population persistence, and how these reductions in sub-lethal performance will affect ecological interactions (e.g., reproduction, disease resistance, predation, and competition) is likely to be important. The slow rate of surface warming in the Southern Ocean [38] means that even a small amount of acclimation can provide species with a significant additional thermal buffer for many decades, long enough for generational plasticity and the possibility of adaptive change, even in Antarctic species [11].

It is therefore not surprising that *O. validus* did not show acclimatisation of acute CT_max_ in the field over the temperature range −1.8 to +1.1 °C. It is probable that the duration of the peak of summer temperatures (1–2 months [24]) or the maximum environmental temperature (+1.1 °C in the field) were not sufficient to trigger whole animal physiological adjustment of CT_max_. The lack of predictable seasonal temperature change in the Southern Ocean could explain the lack of ecologically relevant acclimatisation, the observation that species do not select preferred temperatures, and why there are no observable escape response thresholds [15,16]. However, many other factors, such as tissue energy status, affect CT_max_ (e.g., [39]), and there might be greater seasonal differences in thermal tolerance in primary consumers with seasonally restricted food supplies, than in predators/scavengers which have a more constant food supply [40]. The complexity of the triggers for molecular thermal protective mechanisms has been illustrated for the HSP70 heat shock response in the Antarctic limpet, *N. concinna*. Acute laboratory thermal challenge did not elicit upregulation of HSP70 gene expression until temperatures of 15 °C, whereas limpets in the wild expressed HSP70 at foot temperatures of only 3.3 °C during a natural tidal cycle [41].

This study has highlighted species-specific variation in long-term survival at elevated temperatures, acclimation response and the influence of oxygen on CT_max_ of Antarctic marine ectotherms. The search for mechanisms underlying the evolution of physiological capacity [1,2,42] provides important frameworks against which exceptions can be identified and explanatory hypotheses constructed for future research. Thermal sensitivity is clearly the product of multiple ecological and physiological capacities, and this diversity of response needs further investigation and interpretation to improve our ability to predict future patterns of biodiversity.

## 5. Conclusions

Tests of three of the central paradigms underlying physiological response of Antarctic marine ectotherms to warming highlighted species-specific responses in regard to resilience, plasticity and the effect of oxygen limitation. This further highlights the complexity of thermal physiology and the need for further investigation and interpretation.

## Figures and Tables

**Figure 1 biology-13-00224-f001:**
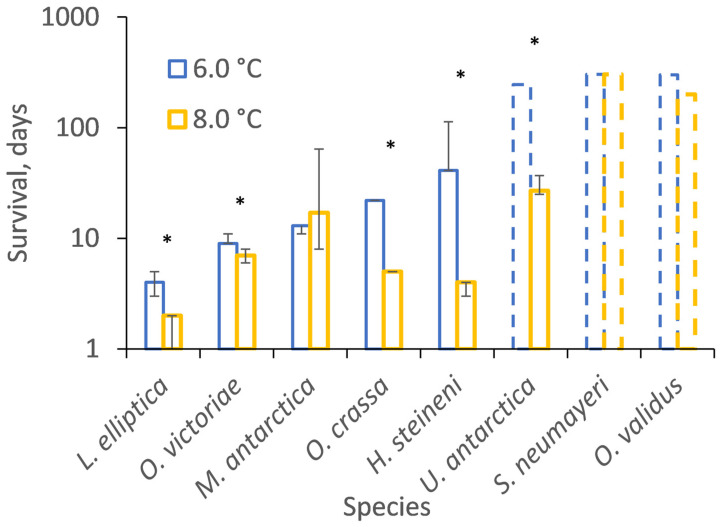
The Kaplan-Meier estimate of median survival duration, in days, after incubation at 6.0 and 8.0 °C. Species ordered from lowest to highest survival. Median ± interquartile range. Hashed lines indicate the duration of experiments for species for which the Kaplan-Meier estimator could not be calculated as most individuals survived until the experiment ended (censored observations), except for *O. validus* at 8.0 °C when the experiment terminated after 200 days due to a heater failure. Missing interquartile ranges indicate that the Kaplan-Meier could not estimate that value (Appendix A). An asterix (*) indicates significantly different survival of that species between incubation temperatures. See Appendix A for the stability of incubation temperatures and Appendix A for Kaplan-Meier estimation figures.

**Figure 2 biology-13-00224-f002:**
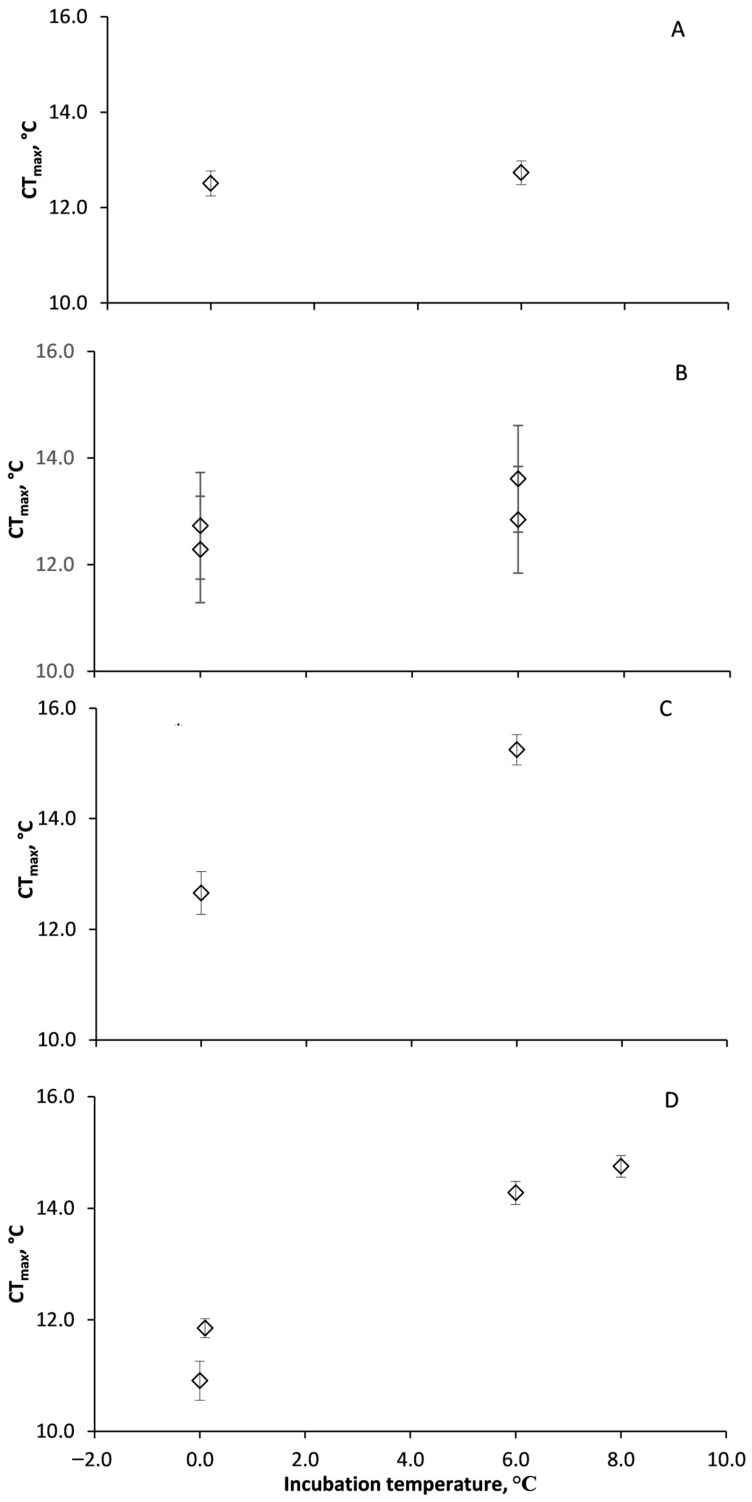
Lethal temperature limits in four Antarctic marine invertebrates incubated in the laboratory at 0.0, 6.0 and 8.0 °C. (**A**) *Urticinopsis antarctica*, (**B**) *Heterocucumis steineni*, (**C**) *Odontaster validus*, and (**D**) *Sterechinus neumayeri*. *U. antarctica* was the only species whose CTmax did not acclimate between control (0.0 °C) and elevated temperatures (6 and 8.0 °C). Where more than one data point is shown for a given temperature, more than one experiment was conducted at different times. *Urticinopsis antarctica*, 0 °C n = 14, 6 °C n = 13, *Heterocucumis steineni*, n = 19 to 24, *Odontaster validus*, 0 °C n = 14, 6 °C n = 19 and *Sterechinus neumayeri*, 0 °C n = 12 and 8, 6 °C n = 21, 8 °C n = 10.

**Figure 3 biology-13-00224-f003:**
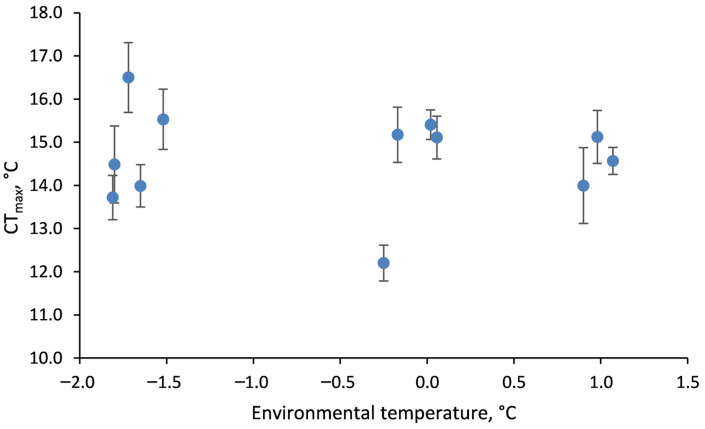
Upper lethal limits of *Odontaster validus* tested directly after collection from the field in different years and seasons. Mean ± 2 SE.

**Figure 4 biology-13-00224-f004:**
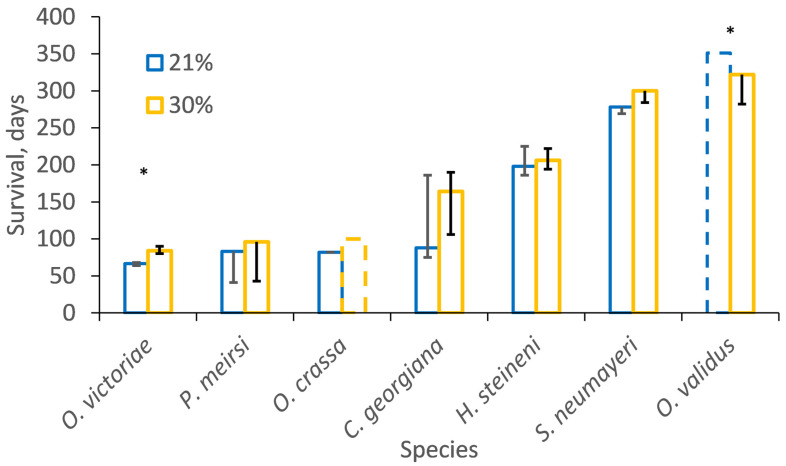
The Kaplan-Meier estimated median survival duration, in days (±interquartile range) that each species survived when incubated under normoxia (21%) or hyperoxia (30%). Hashed lines indicate the duration of experiments for species for which the Kaplan-Meier estimator could not be calculated as most individuals survived until the experiment ended (censored observations). An asterix (*) indicates species with significantly different survivals under different oxygen treatments. For stability of incubation temperature see Table 1 and Appendix A for the Kaplan-Meier estimation figures.

**Table 1 biology-13-00224-t001:** Summary of the conditions (± SE) for temperature–oxygen incubations and a list of the species used in each experiment with sample size per tank in parentheses. A step-change in temperature in the 2014/15 experiment is indicated in parentheses. Species are (A) *Paraceradocus miersii* (30), *Ophionotus victoriae* (30), *Ophiura crassa* (30) and (B) *Sterechinus neumayeri* (10), *Odontaster validus* (10), *Cucumaria georgiana* (10), and *Heterocucumis steineni* (10).

Year	Duration (d)	Tank	Treatment	Temperature (°C)	Oxygen (% O_2_)	Oxygen (mg/L) *
(A)						
2010/11	100	1	Control	0.9 ± 0.05	21.0 ± 0.1	13.9
		2	Normoxic	3.0 ± 0.01	21.0 ± 0.1	13.2
		3	Hyperoxic	3.0 ± 0.02	29.3 ± 0.5	18.3
(B)						
2014/15	351	1	Control	0.1 ± 0.01	21.0 ± 0.1	14.2
		2	Normoxic	8.6 ± 0.5 (10.0 ± 0.01)	21.0 ± 0.3	11.4
		3	Normoxic	8.1 ± 0.2 (9.9 ± 0.01)	21.5 ± 0.5	11.8
		4	Hyperoxic	8.1 ± 0.2 (10.0 ± 0.01)	30.5 ± 0.6	16.8
		5	Hyperoxic	8.0 ± 0.2 (10.0 ± 0.01)	31.3 ± 0.7	17.2

* Mean oxygen concentration (% O_2_) was converted to mg L^−1^ using standard conversion tables (PreSens Precision Sensing GmbH, Regensburg, Germany).

**Table 2 biology-13-00224-t002:** Survival in each incubation experiment.

Species	Incubation	Mortalities	Survivors	% Survival
*S. neumayeri*	6 °C	10	21	68
	8 °C	2	10	83
*O. validus*	6 °C	1	19	95
	8 °C	Technical failure		
*O. victoriae*	6 °C	27	0	0
	8 °C	23	0	0
*O. crassa*	6 °C	24	0	0
	8 °C	25	0	0
*M. antarctica*	6 °C	13	0	0
	8 °C	19	0	0
*U. antarctica*	6 °C	1	13	93
	8 °C	16	0	0
*L. elliptica*	6 °C	19	0	0
	8 °C	15	0	0
*H. steineni*	6 °C	21	4	16
	8 °C	25	0	0
*O. validus*	21%	8	11	58
	30%	12	8	40
*O. crassa*	21%	26	4	13
	30%	4	26	87
*P. miersi*	21%	15	15	50
	30%	15	15	50
*O. victoriae*	21%	30	0	0
	30%	30	0	0
*C. georgiana*	21%	20	0	0
	30%	20	0	0
*S. neumayeri*	21%	18	3	14
	30%	19	0	0
*H. steineni*	21%	20	0	0
	30%	19	1	5

## Data Availability

All data are either published with the manuscript or available on request from the lead author. Morley, S.A., Bates, A.E., Clark, M.S., Fitzcharles, E., Smith, R., Stainthorp, R.E., and Peck, L.S. (2024). Acclimation and acclimatisation of marine ectotherms collected at Rothera Research Station and Scott Base in Antarctica between 2004 and 2015 (Version 1.0) [Data set]. NERC EDS UK Polar Data Centre. https://doi.org/10.5285/60B777B4-0BD6-48C3-A301-C700854FBFA1.

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
