# Peer review of "Testing the Resilience, Physiological Plasticity and Mechanisms Underlying Upper Temperature Limits of Antarctic Marine Ectotherms"

_biology, 2024, doi:10.3390/biology13040224_

Round 1

Reviewer 1 Report

Comments and Suggestions for Authors

Dear Authors,

I have no objection in principle to your manuscript. You have conducted interesting and important research, concerning Antarctic marine fauna. This is particularly important in view of the global climatic changes currently observed.

However, I would recommend some changes to your manuscript.

Lines 106 - 112. This is a piece of text you have left over from the template. It should be deleted.

Line 28. When first mentioned, the name of the species should be written in full. Furthermore, I would recommend that when first mentioned, the author and year should be given. For example, Ophiura crassa Mortensen, 1936. Although, if this is not generally accepted in your field of zoology, you may not add this.

Line 143. Nanochloropsis spp. and Tetraselmis spp. Here spp. should be without italics.

Line 141. I would replace ad libitum with the English word.

Figures 1 and 4, Figures S2 and S3. I would recommend replacing with more contrasting colours. The yellow colour may not be as clear, especially in the printed version of the article.

Table S2. Sterechinus neumayeri should be italicised here.

Line 552. References. Pagothenia borchgrevinki should be italicised.

Author Response

I have no objection in principle to your manuscript. You have conducted interesting and important research, concerning Antarctic marine fauna. This is particularly important in view of the global climatic changes currently observed.

R: we are pleased that the reviewer appreciated our study

However, I would recommend some changes to your manuscript.

Lines 106 - 112. This is a piece of text you have left over from the template. It should be deleted.

R: we apologise. This should clearly have been deleted before submission. We have now deleted this paragraph.

Line 28. When first mentioned, the name of the species should be written in full. Furthermore, I would recommend that when first mentioned, the author and year should be given. For example, Ophiura crassa Mortensen, 1936. Although, if this is not generally accepted in your field of zoology, you may not add this.

R: We have made sure that the species names are mentioned in full at first mention. We have also added the naming authorities for our experimental animals on first mention. We have not done this for species cited from the literature.

Line 143. Nanochloropsis spp. and Tetraselmis spp. Here spp. should be without italics.

R: We thank the reviewer. We have corrected this.

Line 141. I would replace ad libitum with the English word.

R: We have replaced ad libitum with “to excess”

Figures 1 and 4, Figures S2 and S3. I would recommend replacing with more contrasting colours. The yellow colour may not be as clear, especially in the printed version of the article.

R: We have considered this. MDPI is an online journal only and blue and yellow contrast well on screen and are colour blind friendly. Even in print outs the difference between blue and yellow is clear in grayscale. @Editors, if you don’t agree then please let me know and I can recreate the figures

Table S2. Sterechinus neumayeri should be italicised here.

R: Corrected

Line 552. References. Pagothenia borchgrevinki should be italicised.

R: We realise that in copying the references across into the MDPI template the italics were lost from many of the species names in the references. These have been corrected.

Reviewer 2 Report

Comments and Suggestions for Authors

The manuscript presents an overview of NASD research on 10 species of marine invertebrates from Antarctica conducted in 2006–2015 in order to determine the possibility of survival in elevated ambient temperatures and the possibility of adaptation to warming waters. A number of experiments were carried out in controlled conditions (long-term - up to 351 days - rearing) at temperatures increased to 6ï‚°C and 8ï‚°C (i.e. at temperatures from 0 to 4°C above the long-term limits calculated on the basis of previous experiments), as well as in conditions hyperoxia and in normoxia  (long-term incubation under normoxic (21%) and mildly hyperoxic (30%) oxygen saturation). Survival was extrapolated from CTmax estimates using the Kaplan-Meier estimator. The manuscript is concluded with an interesting discussion relating the obtained results to those obtained from studies conducted on natural conditions in waters in the Southern Ocean.

I do not find any significant shortcomings. However, I propose to correct the paragraph in lines 106-112 - in the current version it is a literal copy of the instructions contained in the MDPI text "Microsoft Word template" and to remove "Folmer et al. 1994” on lines 164-165.

Author Response

We thanks the reviewer for their positive assessment. As stated in the response to reviewer 1 we apologise for not deleting the method instructions from the template. This has now been deleted.

Folmer et al. 1994 has been deleted